

# Substitutions into amino acids that are pathogenic in human mitochondrial proteins are more frequent in lineages closely related to human than in distant lineages

Galya V. Klink[1], Andrey V. Golovin[2] and Georgii A. Bazykin[1,3]

[1] Sector of Molecular Evolution, Institute for Information Transmission Problems (Kharkevich Institute) of the Russian Academy of Sciences, Moscow, Russian Federation
[2] Faculty of Bioengineering and Bioinformatics, Lomonosov Moscow State University, Moscow, Russian Federation
[3] Center for Data-Intensive Biomedicine and Biotechnology, Skolkovo Institute of Science and Technology, Skolkovo, Russian Federation

## ABSTRACT

Propensities for different amino acids within a protein site change in the course of evolution, so that an amino acid deleterious in a particular species may be acceptable at the same site in a different species. Here, we study the amino acid-changing variants in human mitochondrial genes, and analyze their occurrence in non-human species. We show that substitutions giving rise to such variants tend to occur in lineages closely related to human more frequently than in more distantly related lineages, indicating that a human variant is more likely to be deleterious in more distant species. Unexpectedly, substitutions giving rise to amino acids that correspond to alleles pathogenic in humans also more frequently occur in more closely related lineages. Therefore, a pathogenic variant still tends to be more acceptable in human mitochondria than a variant that may only be fit after a substantial perturbation of the protein structure.

## INTRODUCTION

Fitness conferred by a particular allele depends on a multitude of factors, both internal to the organism and external to it. Therefore, relative preferences for different alleles change in the course of evolution due to changes in interacting loci or in the environment. In particular, changes in the propensities for different amino acid residues at a particular protein position, or single-position fitness landscape (SPFL, *Bazykin, 2015*), have been detected using multiple approaches (*Bazykin, 2015*; *Storz, 2016*; *Harpak, Bhaskar & Pritchard, 2016*).

One way to observe such changes is by analyzing how amino acid variants (alleles) and substitutions giving rise to them are distributed over the phylogenetic tree. In particular, multiple substitutions giving rise to the same allele, or homoplasies, are more frequent

Corresponding author
Georgii A. Bazykin,
gbazykin@iitp.ru

in closely related species—a pattern expected if frequent homoplasies mark the segments of the phylogenetic tree where the arising allele confers high fitness (*Rogozin et al., 2008*; *Povolotskaya & Kondrashov, 2010*; *Naumenko, Kondrashov & Bazykin, 2012*; *Goldstein et al., 2015*; *Zou & Zhang, 2015*).

Another manifestation of changes in SPFL is the fact that amino acids deleterious and, in particular, pathogenic in humans are often fixed as the wild type in other species. This phenomenon has been termed compensated pathogenic deviations, under the assumption that human pathogenic variants are non-pathogenic in other species due to compensatory (or permissive) changes elsewhere in the genome (*Kondrashov, Sunyaev & Kondrashov, 2002*; *Soylemez & Kondrashov, 2012*; *Jordan et al., 2015*). How often substitutions giving rise to the allele pathogenic in humans occur in non-human species, and how this rate varies with evolutionary distance from the humans, is informative of the mechanism by which acceptability of such substitutions changes in the course of evolution (*Kondrashov, Sunyaev & Kondrashov, 2002*). However, this dynamics has remained controversial. In nuclear genomes, the substitution rate into the allele pathogenic in humans has been found to either stay constant or increase with phylogenetic distance from the human. In earlier analyses, a non-human species was shown to carry an amino acid pathogenic in humans at ~10% of all sites differing between that species and the human, and this value was nearly independent of the phylogenetic distance from the human (*Kondrashov, Sunyaev & Kondrashov, 2002*; *Soylemez & Kondrashov, 2012*). This implied that the rate of substitution to a human pathogenic variant was independent of the degree of relatedness to human. A more recent analysis (*Jordan et al., 2015*), however, has shown that the phylogenetic distance from human to the most closely related species in which the human pathogenic variant is observed is not distributed exponentially as would be expected if the rate of substitution to a human pathogenic variant was the same for all non-human lineages. Instead, that distribution was best represented by a sum of two exponential distributions (*Jordan et al., 2015*), implying that the rate of substitution to a human pathogenic variant increases with phylogenetic distance from the human.

SPFLs of mitochondrial protein-coding genes also change with time (*Goldstein et al., 2015*; *Zou & Zhang, 2015*; *Klink & Bazykin, 2017*), and some of these changes may be due to intragenic or intergenic epistasis (*Ji et al., 2014*; *Xie et al., 2016*). Here, we ask how the fitness conferred by the amino acid variants observed in human mitochondrial proteins, either as benign or damaging, changes with phylogenetic distance from the human. To do that, we use our previously developed approach to study of phylogenetic clustering of homoplasies at individual protein sites (*Klink & Bazykin, 2017*).

## MATERIALS AND METHODS

### Data

Here, we reanalyzed the phylogenetic data on a set of five mitochondrial protein-coding genes in 4,350 species of opisthokonts (*Klink & Bazykin, 2017*), as well as on each of the 12 mitochondrial protein-coding genes in several thousand species of metazoans (*Breen et al., 2012*; *Klink & Bazykin, 2017*). These two datasets strongly overlap (Table S1).

We focused on the amino acids observed in the human mitochondrial genome as reference, polymorphic, or pathogenic variants. A joint alignment of five concatenated mitochondrial genes of opisthokonts and alignments of 12 mitochondrial proteins of metazoans (*Breen et al., 2012*) were obtained as described in *Klink & Bazykin (2017)*. These alignments were used to reconstruct constrained phylogenetic trees, ancestral states and phylogenetic positions of substitutions as in *Klink & Bazykin (2017)*. Briefly, we obtained taxonomy-based trees from the Interactive Tree of Life Project (*Letunic & Bork, 2007*), resolving multifurcations and estimating branch lengths using RAxML (*Stamatakis, 2014*). As the reference human allele, we used the revised Cambridge Reference Sequence (rCRS) of the human mitochondrial DNA (*Andrews et al., 1999*). As non-reference alleles, we used amino acid changing variants from "mtDNA Coding Region & RNA Sequence Variants" section of the MITOMAP database (*Lott et al., 2013*). As pathogenic alleles, we used amino acid changing variants with "reported" (i.e., considered by one or more publication as possibly pathogenic) or "confirmed" (i.e., supported by at least two independent laboratories as pathogenic) status from the "Reported Mitochondrial DNA Base Substitution Diseases: Coding and Control Region Point Mutations" section of MITOMAP.

## Clustering of substitutions giving rise to the human amino acid around the human branch

For each amino acid site in a protein, we considered those amino acid variants that (i) constitute the reference allele in humans and had arisen in the human lineage at some point during its evolution, or (ii) had originated in humans as a derived polymorphic allele, or (iii) are annotated in humans as pathogenic alleles. For further consideration, we retained only such alleles from each class for which at least one homoplasic (i.e., giving rise to the same allele by way of parallelism, convergence, or reversal) and at least one divergent (i.e., giving rise to a different allele) substitution from the same ancestral variant was observed at this site elsewhere on the phylogeny outside of the human lineage. Substitutions, including reversals, that occurred anywhere on the path between the tree root and *H. sapiens* were excluded. While the homoplasic and divergent substitutions had to derive from the same ancestral variant, it could be either the same or a different variant than that ancestral to the variant observed in human.

For each such allele, we compared the phylogenetic distances between human and positions of homoplasic substitutions with the distances between human and positions of divergent substitutions, using a previously described procedure which controls for the differences in SPFLs between sites or in mutational probabilities of different substitutions (*Klink & Bazykin, 2017*). Briefly, for each ancestral amino acid at each site, we subsampled equal numbers of homoplasic substitutions to human (reference, non-reference or pathogenic) amino acids and of divergent substitutions to non-human amino acids. The subsample size was $\min(N_{homo}, N_{diverg})$, where $N_{homo}$ and $N_{diverg}$ are the numbers of homoplasic and divergent substitutions originating from this amino acid at this site. We repeated this procedure for all ancestral amino acids at all sites, thus obtaining two equal-sized subsamples of homoplasic and divergent substitutions, and measured all distances

in these resulting subsamples. We then pooled these values across all considered sites and groups of derived alleles. Next, we categorized them by the phylogenetic distance between the human and the position of the substitution, and calculated, for each bin of the phylogenetic distances, the ratio of the numbers of homoplasic (H) and divergent (D) substitutions (H/D). To obtain the mean values and 95% confidence intervals for the H/D statistic, we bootstrapped sites in 1,000 replicates, each time repeating the entire resampling procedure. As a control, we performed the same analyses using instead of the human variant a random non-human amino acid among those observed at this site, or using data obtained by simulating the evolution at each site along the same phylogeny and with gene-specific GTR+Gamma amino acid substitution matrices using the evolver program of the PAML package (*Yang, 1997*) as in *Klink & Bazykin (2017)*.

### Molecular dynamics simulation of single point mutations in position 91 of COX3

The amino acid at position 91 of COX3 is adjacent to the pore in the complex IV of the respiratory chain which is thought to be required for the transport of oxygen. Therefore, estimation of the effect of the mutation in this position requires a complete description of the cytochrome c oxidase complex in membrane environment. We applied coarse grain description with martini forcefield (*Marrink et al., 2007*). Human cytochrome C oxidase was modelled with homology modelling approach using Modeller 9.18 (*Eswar et al., 2006*). The membrane environment was rebuild from a random position of lipids and restrained protein structure as in MemProt MD database (*Stansfeld et al., 2015*). Each mutant was subjected to 0.5 mks simulation with two replicas in GROMACS 2016 (*Abraham et al., 2015*) with time step 10fs with PME electrostatics (*Wennberg et al., 2015*). Simulation results were analyzed with MDAnalysis Python module (*Michaud-Agrawal et al., 2011*). Water molecules within the pore were counted in a cylindrical selection with radius 2 nm, height 5 nm and center position determined dynamically as the center of mass of the two helices of the dimer harboring mutations. Counts of water molecules and their standard deviations were estimated from the last 100 ns of the trajectory.

## RESULTS

### Substitutions giving rise to reference human amino acids are more frequent in species closely related to human

The phylogenetic distribution of homoplasic (parallel, convergent or reversing) substitutions is relevant for understanding which amino acids are permitted at a particular species, and which are selected against. In particular, an excess of such substitutions giving rise to a particular allele in closely related lineages implies that the fitness conferred by this allele in these species is higher than in other species. Here, using phylogenies reconstructed from two datasets of mitochondrial protein coding genes obtained earlier (*Klink & Bazykin, 2017*), we asked how homoplasic substitutions giving rise to the human variant are positioned phylogenetically relative to the human branch, compared with other (divergent) substitutions.

The reference human allele is also observed, on average, in over half of other considered species (71.2% of species of opisthokonts for the 5-genes dataset, and 60% of all species of metazoans for the 12-genes dataset). In 7.1% (10%) of these species, this allele did not share common ancestry with the human, but instead originated independently in an average of 30 (21) independent homoplasic substitutions per site (Table S2 and Figs. S1–S2).

We asked whether the phylogenetic positions of homoplasic substitutions giving rise to the human allele are biased towards the human branch, compared to the positions of divergent substitutions giving rise to other alleles. This analysis controls for the biases associated with pooling sites and amino acid variants (see 'Materials and Methods'). Although it does not allow us to analyze very conserved sites, most sites were sufficiently variable for this and all subsequent analyses (Tables S1–S3).

In most proteins, the mean phylogenetic distances from human to the branches at which substitutions to the human reference amino acid occurred independently due to a homoplasy were ∼10% shorter than to the branches at which substitutions to another amino acid occurred (Fig. 1A; Fig. S3). No such decrease was observed for a random amino acid among those that were observed at this site in non-human species, or in simulated data (Fig. 1A; Fig. S3). The number of homoplasic substitutions giving rise to the human reference amino acid relative to the number of divergent substitutions towards non-human amino acids (H/D ratio) uniformly decreases with the evolutionary distance from the human (Fig. 1B; Fig. S4). At most genes, the relative number of homoplasic substitutions giving rise to the human allele drops 1.5–3-fold with phylogenetic distance from the human branch (Fig. 1B; Fig. S4), implying that such substitutions preferentially occur in lineages more closely related to humans.

The excess of homoplasic changes to the human reference amino acid at small phylogenetic distances from human is not an artefact of differences in mutation rates between amino acids in distinct clades, since these rates are similar in all considered species and cannot lead to such clustering (*Klink & Bazykin, 2017*). It is also not an artefact of differences in codon usage bias between species, as it was still observed when we considered only "accessible" amino acid pairs where the derived amino acid could be reached through a single nucleotide substitution from any codon of the ancestral amino acid (Fig. S3).

## Substitutions to amino acids corresponding to variant alleles at human polymorphic sites more frequently occur in species closely related to humans

Next, we considered human SNPs in mitochondrial proteins in the MITOMAP database. These SNPs are representative of human polymorphism across multiple populations, although as any such database, it preferentially represents variants that are more frequent. We analyzed the phylogenetic distribution of substitutions in non-human species giving rise to the amino acid that is also observed as the non-reference (usually minor) allele in humans (Table S3).

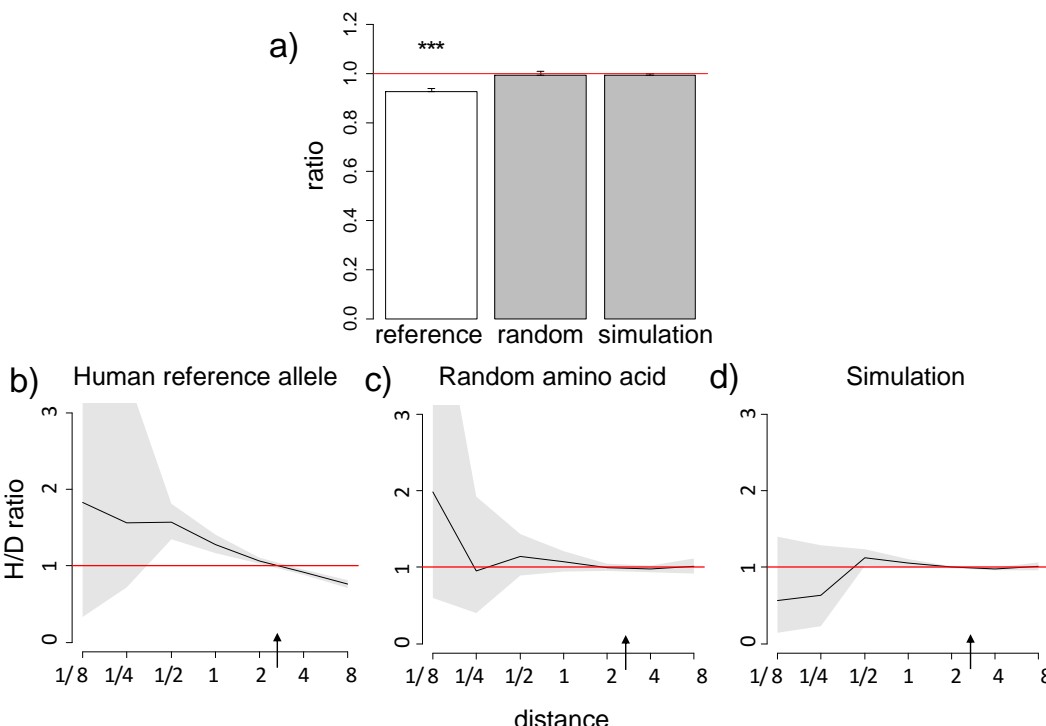

**Figure 1   Homoplasic substitutions to the human reference amino acid tend to occur in species closely related to human.** A reduced ratio of the phylogenetic distances between the human branch and substitutions to the considered amino acid vs. to other amino acids at the same site (A) and higher fraction of homoplasic substitutions to the human reference amino acid, compared with random amino acids that had independently originated at this site (H/D ratio) in species closely related to human (B) are observed for the human reference amino acid, but not for a random allele observed at this site (A, C) or a simulated allele (A, D), in the 4,350-species opisthokonts phylogeny. (A) Ratios < 1 imply that the considered allele arises independently closer at the phylogeny to humans than other alleles. The bar height and the error bars represent respectively the median and the 95% confidence intervals obtained from 1,000 bootstrap replicates, and asterisks show the significance of difference from the one-to-one ratio (*, $P < 0.05$; **, $P < 0.01$; ***, $P < 0.001$). Reference, human reference allele; random, a random non-human amino acid among those that were present in the site; simulation, human allele in simulated data. (B, C, D) Horizontal axis, distance between branches carrying the substitutions and the human branch, measured in numbers of amino acid substitutions per site, split into bins by $\log_2$(distance). Vertical axis, H/D ratios for substitutions at this distance. Black line, mean; grey confidence band, 95% confidence interval obtained from 1,000 bootstrapping replicates. The red line shows the expected H/D ratio of 1. Arrows represent the distance between human and *Drosophila*.

Similarly to the human reference amino acid variant, the homoplasic substitutions giving rise to non-reference alleles were clustered on the phylogeny near humans, compared to the divergent substitutions giving rise to a variant never observed in humans (Fig. 2; Figs. S5–S6). Again, the mean phylogenetic distance from human to a substitution giving rise to the human non-reference amino acid was ~10% lower than to other substitutions (Fig. 2A; Fig. S5), and the density of homoplasic substitutions giving rise to such an allele dropped significantly with phylogenetic distance from the human branch ($p < 0.001$, Fig. 2B; Fig. S6). As before, this clustering was also observed if only accessible pairs of

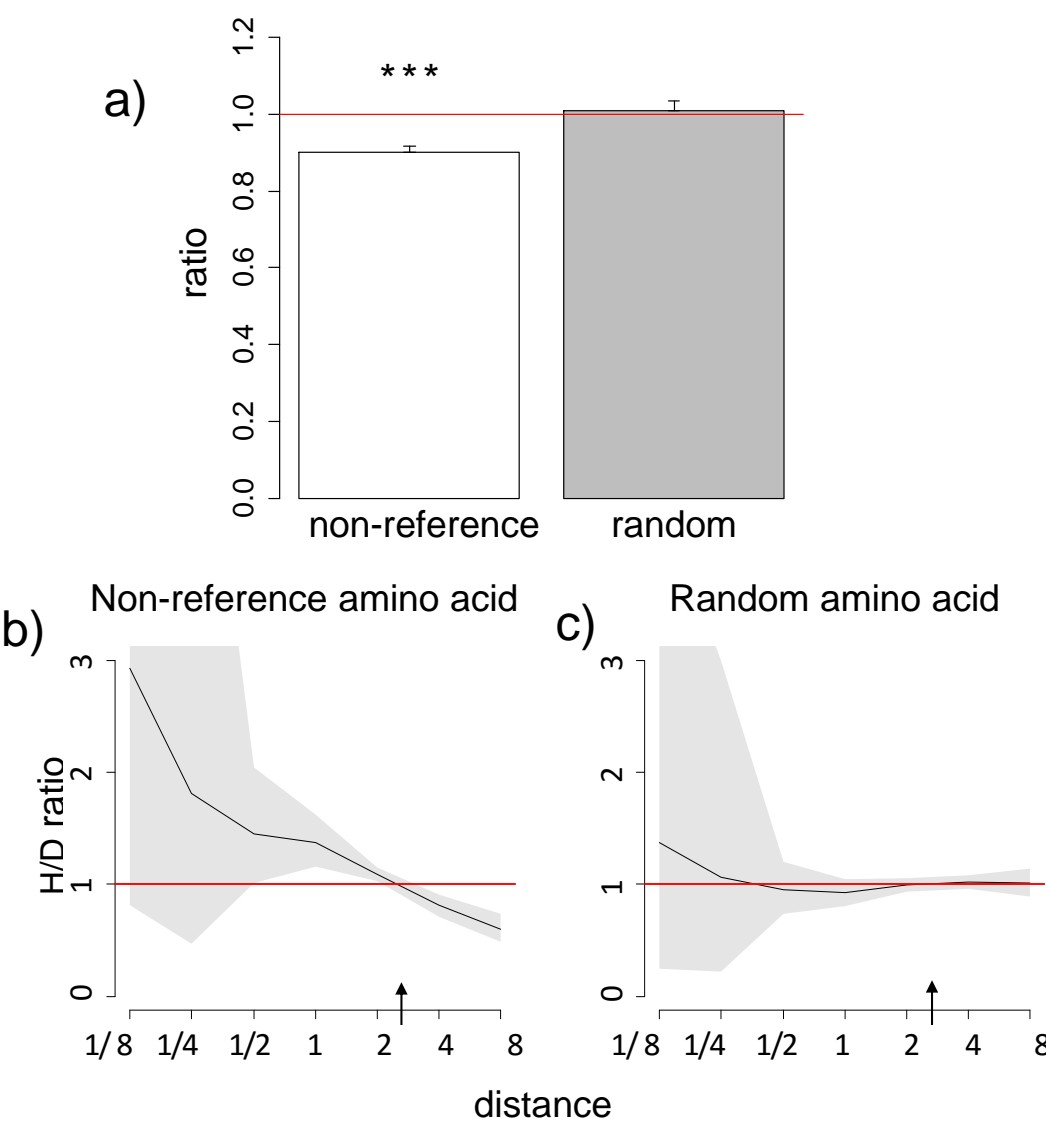

**Figure 2 Homoplasic substitutions to the human non-reference amino acid tend to occur in species closely related to human.** A reduced ratio of the phylogenetic distances between the human branch and substitutions to the considered amino acid vs. to other amino acids at the same site (A) and a higher H/D ratio in species closely related to human (B) are observed for the human non-reference amino acid, but not for a random allele observed at this site (A, C), in the 4,350-species opisthokonts phylogeny. Notations same as in Figs. 1 and 2.

amino acids were considered, while no systematic differences were observed for random amino acids (Fig. S5).

## Substitutions to amino acids corresponding to human pathogenic variants more frequently arise in species closely related to humans

Finally, we considered human alleles annotated as disease-causing in the MITOMAP database (*Lott et al., 2013*). Since only a handful of mutations is thus annotated in each gene (Table S4), the variance in the estimates of the H/D ratio is, as expected, large. Still,

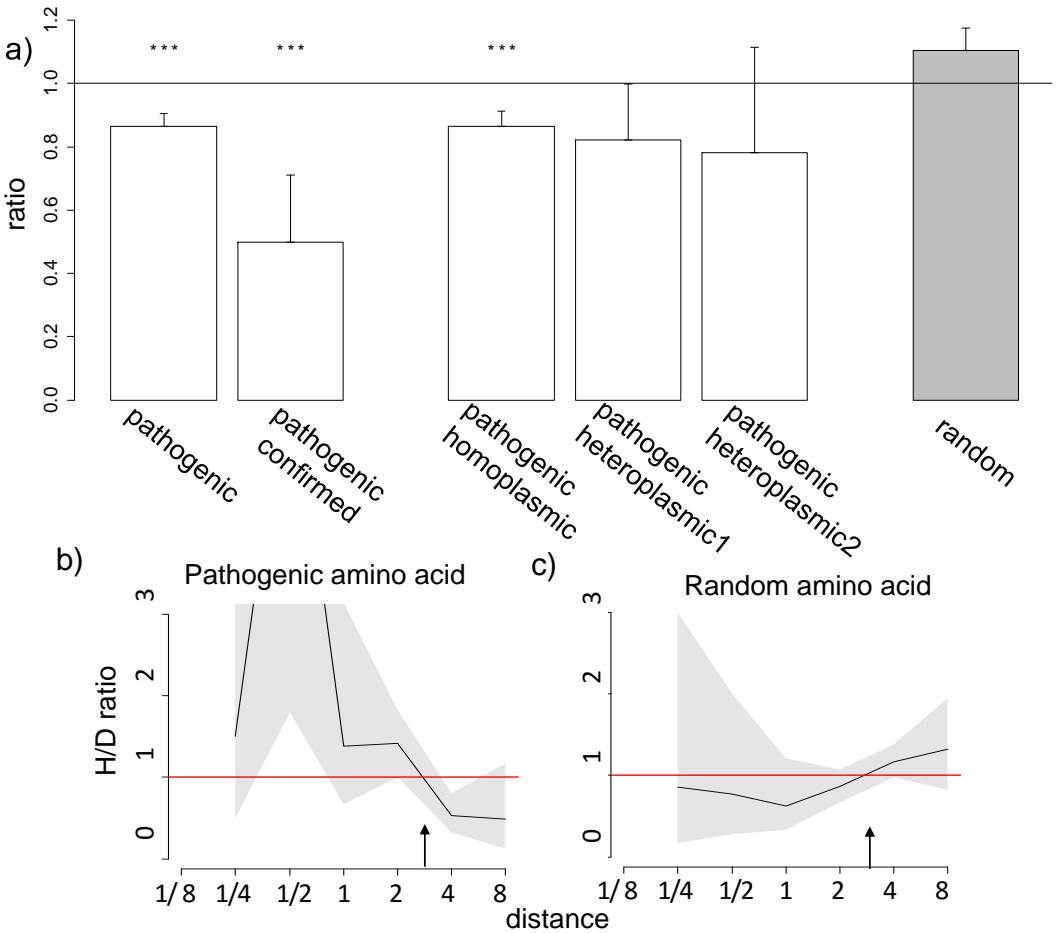

**Figure 3   Homoplasic substitutions to the human pathogenic amino acid tend to occur in species closely related to human.** (A) A reduced ratio of the phylogenetic distances between the human branch and substitutions to the considered amino acid vs. to other amino acids at the same site are observed for all (pathogenic), confirmed (confirmed) and homoplasmic (homoplasmic) human pathogenic amino acid, but not for a random allele observed at this site (random) or for pathogenic amino acids that have been seen only in heteroplasmic states (heteroplasmic1,2), in the 4,350-species opisthokonts phylogeny; heteroplasmic1–mutations that have been only observed as heteroplasmic; heteroplasmic2–mutations that have been only observed as heteroplasmic with more than one reference in MITOMAP. Other notations same as in Figs. 1 and 2. (B, C) A higher H/D ratio in species closely related to human are observed for human pathogenic amino acid, but not for a random allele observed at this site in the 4,350-species opisthokonts phylogeny. Notations same as in Figs. 1 and 2.

in the opisthokont dataset (Fig. 3), as well as in five of the twelve genes of the metazoan dataset (Figs. S7–S8), the human pathogenic variant also arose independently more frequently in the phylogenetic vicinity of humans. The opposite pattern, i.e., biased occurrence of the human pathogenic variant in phylogenetically remote species, has not been observed in any of the genes. This trend was even stronger for the six pathogenic mutations confirmed by two or more independent studies ("confirmed" status in MITOMAP; Fig. 3A). It was observed for the mutations that had been observed as homoplasmic, but not for the mutations that had always only been seen as heteroplasmic

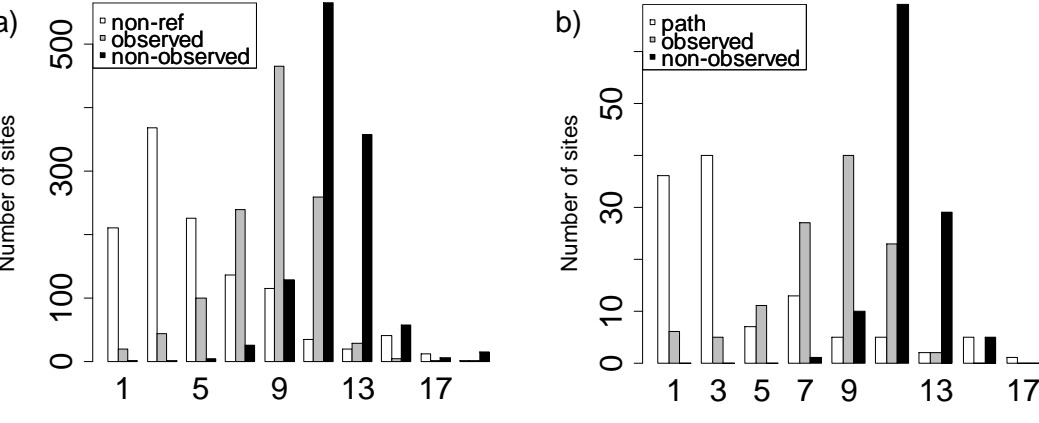

**Figure 4  Miyata distances between the reference and non-reference human alleles.** Distributions of ranks of Miyata distances between the reference human allele and the non-reference (A) or pathogenic (B) allele (white), compared with other alleles that were (gray) or were not (black) observed at the same site. For each site with known polymorphisms or pathogenic mutations, we ranked all amino acids by Miyata distance from reference human allele, and then obtained distance rank for pathogenic (or non-reference) human variant, mean rank for amino acids that occurred in a site but was not observed in human and mean rank for rest amino acids.

(i.e., that were never fixed within an individual) and therefore could confer complete loss of function (Fig. 3A). As before, this result is not due to preferential usage of codons more likely to mutate into the human variant in species closely related to human (Fig. S7).

## Human pathogenic variants are biochemically similar to normal human variants

To understand what drives the substitutions to the human variant, either normal or pathogenic, in species closely related to humans, we analyzed the identity of these variants. Both normal (Fig. 4A) and pathogenic (Fig. 4B) human amino acids were more similar in their biochemical properties according to the Miyata matrix (*Miyata, Miyazawa & Yasunaga, 1979*) to the human reference variant than amino acids observed in non-human species. In turn, amino acids observed in non-human species were more similar to the human reference variant than amino acids never observed at this site in any species.

## Individual mutations

To illustrate the observed phylogenetic clustering, we schematically plotted the distribution over the opisthokont phylogeny of substitutions at the six amino acid sites that carry pathogenic mutations with ''confirmed'' status. Visual inspection of these plots confirms that the substitutions giving rise to pathogenic alleles tend to be clustered in the vicinity of the human, compared to other substitutions of the same ancestral amino acids (Fig. S9). For the metazoan dataset, we also plot three select individual amino acid sites with known pathogenic mutations which are considered below. In all these sites, a higher density of substitutions to a particular amino acid in vicinity of the human branch may imply that its relative fitness is higher in species closely related to human.
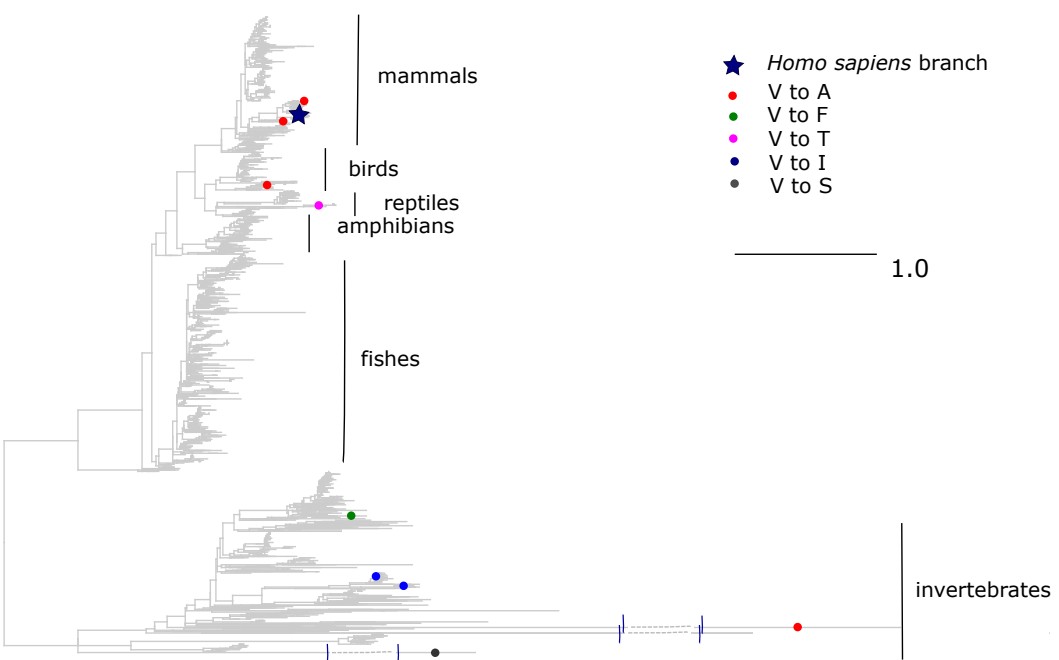

**Figure 5** **Substitutions in site 113 of ND1.** Blue star is *H. sapiens* branch; red dots are substitutions of valine to alanine, which is pathogenic in human and dots of other colors are substitutions to other amino acids. Phylogenetic distances are measured in numbers of amino acid substitutions per site. The branches indicated with the blue lines are shortened approximately by two distance units.

The V → A mutation at ND1 site 113 has been reported to cause bipolar disorder, and decreases the mitochondrial membrane potential and reduces ND1 activity in experiments (*Munakata et al., 2004*). According to the mtDB database (*Ingman & Gyllensten, 2006*), the A allele persists in human population at 0.5% frequency. However, we observed that the same allele originated independently in three clades of vertebrates: old World monkeys (Cercopithecidae), flying lemurs (Cynocephalidae) and turtles (Geoemydidae), while most of the other substitutions of V at this site occurred in invertebrates (Fig. 5). As a result, the mean phylogenetic distance between human and the parallel V → A substitutions is 2.35 (median 0.75), while it is 4.12 (median 4.6) for substitutions of V to other amino acids.

The V → A mutation at COX3 site 91 has been reported to cause Leigh disease (*Mkaouar-Rebai et al., 2011*). In metazoans, A allele at this site has originated independently seven times, including six times from V and once from I. All but one of these substitutions occurred in mammals, while tens of substitutions of V and I giving rise to other amino acids occurred throughout metazoans (Fig. 6). As a result, the mean phylogenetic distance from human to V → A substitutions was 0.6 (median 0.7), while the distance to other mutations from V was 1.8 (median 2.1); the corresponding numbers for I were 0.6 (median 0.6) and 2.7 (median 2.2).

To better understand the possible reasons for the unexpected pattern of clustering of the deleterious variant in the phylogenetic vicinity of human, we used molecular

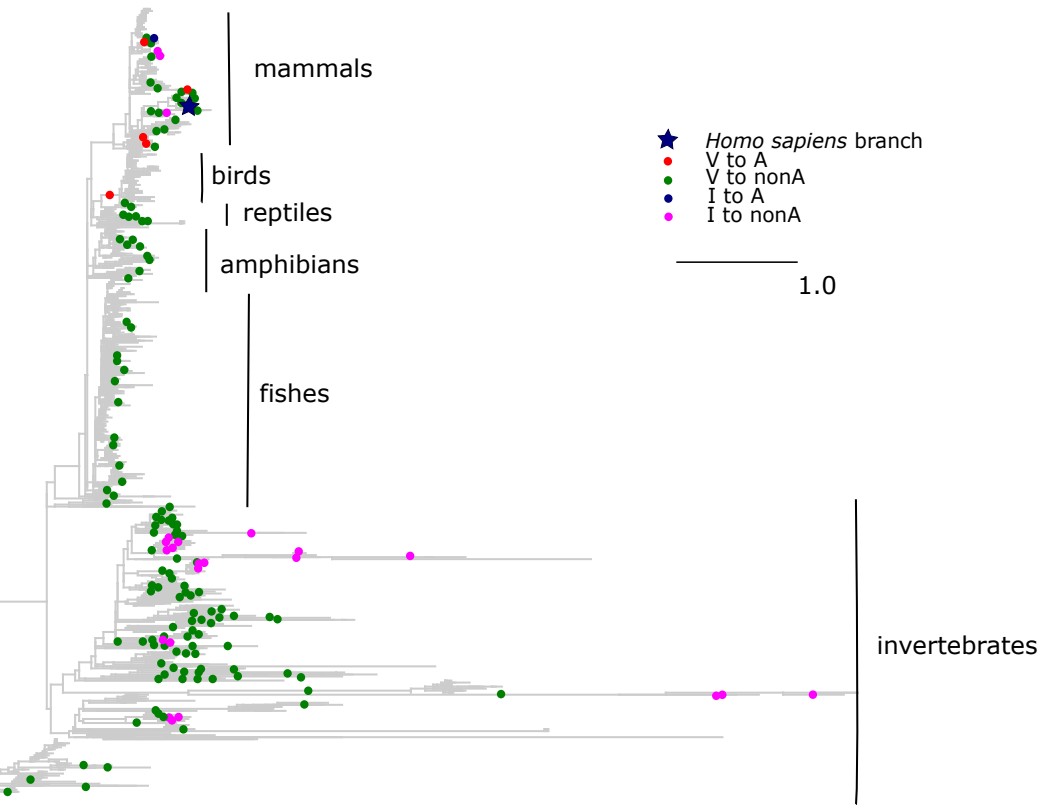

**Figure 6** **Substitutions in site 91 of COX3.** Blue star is *H. sapiens* branch; red dots are substitutions from valine to alanine which is pathogenic in human, blue dot is substitution from isoleucine to alanine, and dots of other colors are substitutions to other amino acids. Phylogenetic distances are measured in numbers of amino acid substitutions per site.

dynamics simulations to predict the effect of each mutation on the structure and function of the human protein. As site 91 is positioned within the wall of a pore that is thought to be a channel for oxygen transport (*Shinzawa-Itoh et al., 2007*), we estimated the pore size that would correspond to each amino acid that occured at this site elswhere on the phylogeny if it arose in the mammalian context. All amino acids led to an increase of the pore size, compared to the normal V allele. Such an increase is expected to permit water molecules to enter the pore, and to impede or prevent oxygen transport. Among the eight observed amino acids, the human pathogenic variant A alters the pore size to the smallest extent, while variants observed in other species substantially increase it, potentially interfering with function (Fig. 7).

Finally, the I → V mutation at ND6 site 33 has been reported to cause type two diabetes (*Tawata et al., 2000*) and has population frequency of 0.1% according to the mtDB. However, this substitution has occurred repeatedly in parallel in mammals and amphibians, while other substitutions of I were frequent in invertebrates (Fig. 8). The mean phylogenetic distance from human to parallel substitutions to V was 2.4 (median 1.5), while it was as high as 12.9 (median 14.8) for other amino acids.
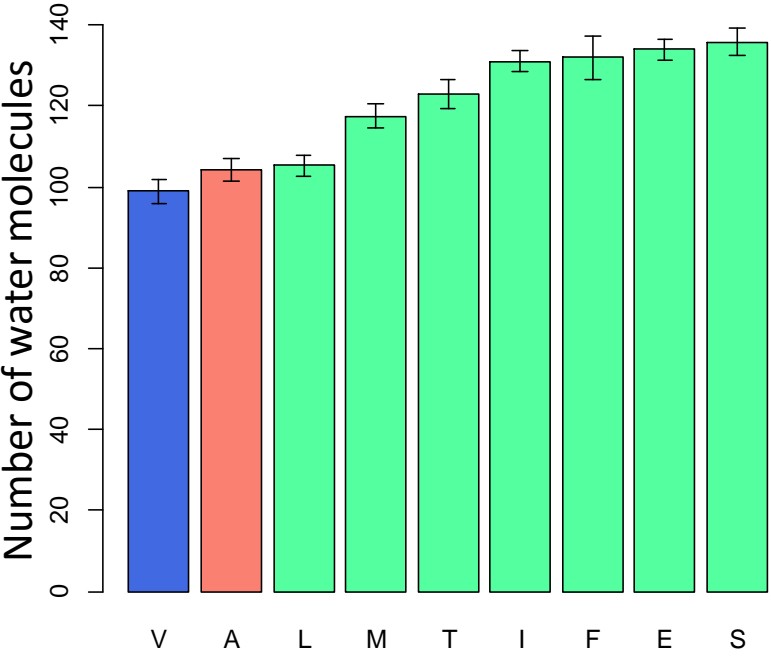

**Figure 7** **Modelled counts of water molecules in the pore of the human mitochondrial cytochrome c oxidase harboring different mutations in position 91 of COX3.** A higher number of molecules probably impedes or prevents oxygen transport. Blue, human reference amino acid (V); red, human pathogenic amino acid (A); green, amino acids observed in non-human species. The bar height and the error bars represent respectively the mean and the standard error of the mean.

## DISCUSSION

A variant deleterious in human may be fixed in a non-human species, and sometimes this can be explained by compensatory or permissive mutations elsewhere in the genome (*Kondrashov, Sunyaev & Kondrashov, 2002*; *Kern &Kondrashov, 2004*; *Jordan et al., 2015*). Here, we reveal the opposite facet of the same phenomenon: a variant that is fixed or polymorphic in human may be deleterious in a non-human species.

Indeed, we find that substitutions giving rise to the human allele occur in species that are more closely related to *H. sapiens* than species in which substitutions to other amino acid occur. While artefactual evidence for excess of parallel substitutions between closely related species may arise from discordance between gene trees and species trees (*Mendes, Hahn & Hahn, 2016*), it is unlikely that it causes the observed signal in our analysis. For reference alleles, we have previously shown that phylogenetic clustering in mitochondrial proteins is not due to tree reconstruction errors (*Klink & Bazykin, 2017*), and mitochondrial genomes do not recombine, which makes other causes of discordance unlikely. For variants polymorphic in humans, artefactual evidence for parallelism could theoretically also arise from a variant that was polymorphic in the last common ancestor of human and another species such as chimpanzee, was subsequently fixed in this other species, and survived as polymorphism in the human lineage until today. However, the last common ancestor of human mitochondrial lineages probably lived no longer than

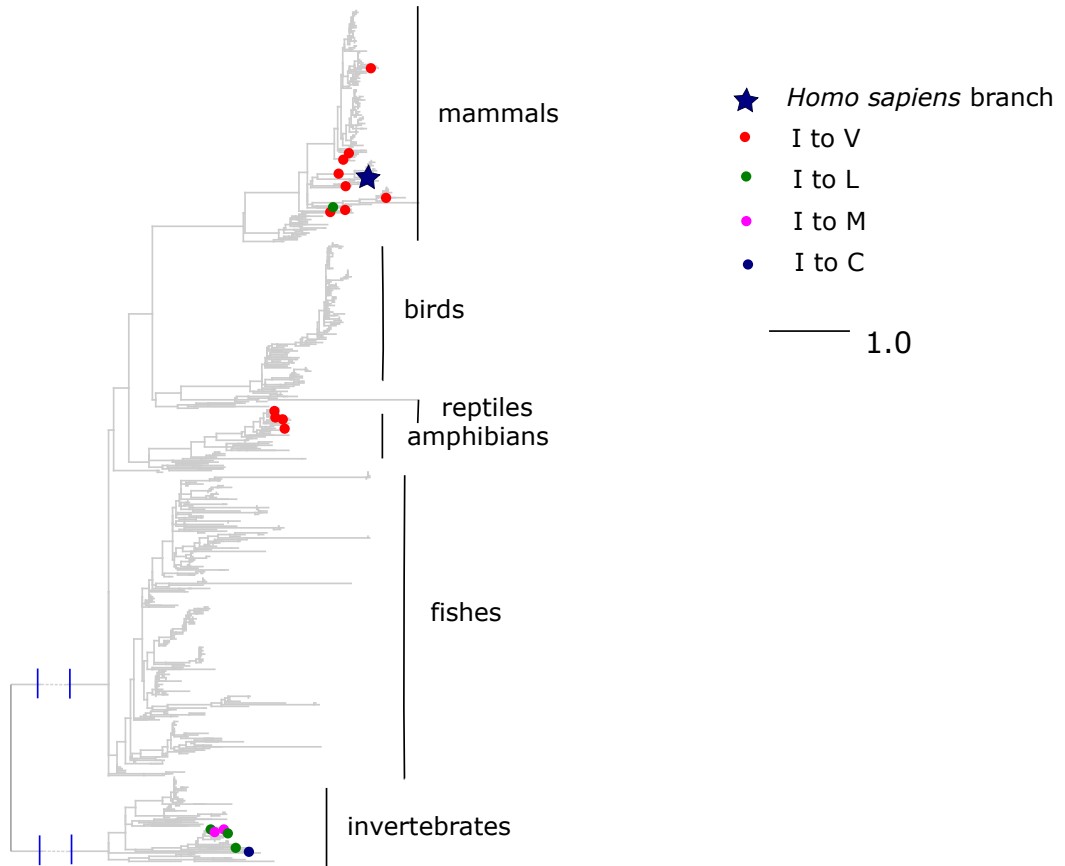

**Figure 8** **Substitutions in site 33 of ND6.** Blue star is *H. sapiens* branch; red dots are substitutions of isoleucine to valine, which is pathogenic in human and dots of other colors are substitutions to other amino acids. Phylogenetic distances are measured in numbers of amino acid substitutions per site. The branches indicated with the blue lines are shortened approximately by three distance units.

148 thousand years ago (*Poznik et al., 2013*), which is much more recent than the time of human-chimpanzee divergence (ca. 6–8 million years ago, *Langergraber et al., 2012*; *Amster & Sella, 2016*), also excluding this option.

Therefore, the observed phylogenetic clustering is real. The rate of a substitution is indicative of the selection coefficient associated with it (*Kimura, 1983*). Thus, this implies that the fitness conferred by the human amino acid relative to that conferred by other amino acids declines with phylogenetic distance from human.

Arguably, one would expect the opposite pattern in the variants pathogenic for humans. Indeed, it is likely that the majority of such variants are also deleterious in species related to humans, while changes in the genomic context or environment in more distantly related species may make these variants tolerable. For nuclear proteins, the distribution of evolutionary distances to the phylogenetically nearest occurrence of a human pathogenic variant has been shown to deviate from the exponential distribution, which would have been expected if such variants were becoming acceptable in the

course of evolution at a uniform rate. On this ground, it has been inferred that this rate increases with decreased relatedness, and that this process involves on average one compensatory (or permissive) substitution in addition to the considered substitution (*Jordan et al., 2015*). If so, a more distantly related species would experience a substitution to a pathogenic allele with a higher probability than a closely related species, as the compensatory substitution is unlikely to have occurred in close relatives. Opposite to this expectation, we observe the pattern similar to that for the non-pathogenic variants: substitutions to the amino acid variants pathogenic for humans are also relatively more likely to occur independently as a homoplasy in species closely related to *H. sapiens* than in more distantly related species.

The similarity between the phylogenetic distribution of the benign and pathogenic variants could be due to some of the benign mutations being incorrectly annotated as pathogenic (*Lek et al., 2016*). However, none of the six mutations with "confirmed" status in Mitomap are present in the mtDB database (*Ingman & Gyllensten, 2006*) among the 2,704 sequences from different human populations, suggesting that these mutations are indeed damaging, while they demonstrate a pronounced clustering (Fig. 3B).

Therefore, the substitutions into alleles pathogenic in humans are indeed more likely to occur in closer human relatives. The difference from the previous results, which suggested no dependence on phylogenetic distance (*Kondrashov, Sunyaev & Kondrashov, 2002*; *Soylemez & Kondrashov, 2012*) or higher prevalence in more distantly related species (*Jordan et al., 2015*; *Harpak, Bhaskar & Pritchard, 2016*) could have several explanations. First, it could be due to differences in the analyzed set of genes (nuclear vs. mitochondrial), or due to a larger range of phylogenetic distances considered here. Second, as our analysis requires several independent substitutions into the pathogenic allele, our sample is biased towards sites of lower conservation. Still, we observed no dependence of the phylogenetic non-uniformity of homoplasies on conservation (*Klink & Bazykin, 2017*, Fig. S3).

Third, the mutations analyzed here could be less deleterious than those considered previously which largely involved complete loss of function. For loss of function mutations, the associated decline in fitness should arguably be the same in all species; therefore, no phylogenetic non-uniformities are expected. As phenotypic effects of pathogenic mutations in mitochondrial-encoded proteins depend on the proportion of mutated mitochondrial DNA in cells (*Wallace, 1992*), loss of function mutations in homoplasmy are likely to be lethal, and thus we only expect to observe loss of function mutations in heteroplasmy. This can explain the fact that many mitochondrial DNA diseases are heteroplasmic (*Chan, 2006*). Indeed, in the considered dataset, all nonsense mutations were heteroplasmic. By contrast, mutations that cause an incomplete loss of function, and therefore that can confer different fitness in different species, may be homoplasmic. In line with our expectations, we observe the phylogenetic clustering for those mutations that have been observed as homoplasmic; by contrast, for mutations that were only observed in heteroplasmy, no phylogenetic clustering was observed (Fig. 3A). The finding that presumably loss of function mutations have a uniform phylogenetic distribution is consistent with our model.

Why does the fitness conferred by the human pathogenic variant appear to be the highest in our closest relatives? Our analysis considers relative fitness, as assessed by the frequency of substitution into a particular allele relative to substitutions into other alleles. Therefore, it can be affected by changes in the fitness conferred by other alleles rather than the considered one. Consideration of biochemical similarities of amino acid variants helps explain why human-pathogenic variants can still be more likely to occur in species closely related to humans. We find that despite their pathogenicity, the human pathogenic variants are on average more biochemically similar to the major human allele than other amino acids that were observed at this site in non-human species. Therefore, in the context of the human genome, the annotated human pathogenic variant probably disrupts the protein structure less, and is therefore less deleterious, than an "alien" non-human variant. Conversely, many alien variants that are not observed in humans are likely to be even more deleterious in human than the annotated pathogenic allele, perhaps lethal, while they confer high fitness in the genomic context of their own species.

Amino acid at position 91 of the COX3 protein provides a case in point. When the mammalian protein structure is used for modelling, we predict that the human pathogenic variant disrupts structure to a smaller extent than each of the seven variants found in reference genomes of other species. Therefore, the reduction in the prevalence of the human pathogenic variant with phylogenetic distance may be due to an increased prevalence (perhaps due to increased fitness) of other variants unfit in humans, rather than a decreased fitness of the known human pathogenic variant.

In summary, we have shown that substitutions to all types of alleles, including pathogenic, that occur in human mitochondrial protein-coding genes more frequently occur independently in species more closely related to *H. sapiens*. Such a decline in occurrence of human amino acids with phylogenetic distance from human is probably due to a higher similarity of the sequence and/or environmental context in more closely related species.

More generally, it is broadly accepted that the occurrence of a mutation in another species is an important predictor of its pathogenicity in humans (*Adzhubei et al., 2010*; *Kumar et al., 2011*), but it is important to account for the degree of relatedness of the considered species to human. Our observation that human-pathogenic alleles are under-represented in more distantly related, rather than in more closely related, species shows that the direction of the association between relatedness and prediction of pathogenicity can be counterintuitive.

## CONCLUSIONS

Using phylogenetic clustering of substitutions to amino acids that represent human (reference, polymorphic or pathogenic) variants in mitochondrial proteins, we suggest that the fitness conferred by such variants often differs between species, and is higher in species that are more closely related to human. For human pathogenic variants, that could be explained by the fact that most known pathogenic mutations in human mitochondria do not lead to complete loss of function, while the fitness associated with them may become even lower in more distantly related species.

## ACKNOWLEDGEMENTS

We thank Shamil Sunyaev, Alexey Kondrashov, Dmitry Pervouchine and Vladimir Seplyarskiy for valuable comments.

### Funding

This work was supported by the Russian Science Foundation (grant number 14-50-00150). Modeling of the structure of COX3 protein has been supported by the Russian Science Foundation (grant number 14-50-00029). The funders had no role in study design, data collection and analysis, decision to publish, or preparation of the manuscript.

### Grant Disclosures

The following grant information was disclosed by the authors:
Russian Science Foundation: 14-50-00150, 14-50-00029.

### Competing Interests

The authors declare there are no competing interests.

### Author Contributions

- Galya V. Klink analyzed the data, contributed reagents/materials/analysis tools, wrote the paper, prepared figures and/or tables.
- Andrey V. Golovin modelling of the structure of COX3 protein.
- Georgii A. Bazykin conceived/designed the study, contributed reagents/materials/analysis tools, wrote the paper, reviewed drafts of the paper.

### Data Availability

   The descriptions of the scripts and raw data are included in the Supplemental Files.

### Supplemental Information

Supplemental information for this article can be found online at http://dx.doi.org/10.7717/peerj.4143#supplemental-information.

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
