# Peer review of "Substitutions into amino acids that are pathogenic in human mitochondrial proteins are more frequent in lineages closely related to human than in distant lineages"

_PeerJ, doi:10.7717/peerj.4143_

## Round 0.1 · original submission · Major Revisions

· Academic Editor

Major Revisions

Both reviewers have a number of specific comments that you should address before publication.

Reviewer 1 ·

Basic reporting

This manuscript demonstrates that alleles known to be pathogenic in humans arise more frequently in lineages closely related to humans. The work laid out here is interesting, though I have a few concerns that I would like to see addressed. Overall, this manuscript is well written and I have included some minor suggestions to improve clarity in my line by line comments. The introduction and the discussion sections would benefit from further discussion of how this study compares to others. Please see the first of my general comments when addressing these comparisons. Some of the figures can be hard to read due to the large phylogenies being shown. I suggest collapsing clades that do not contain relevant information. The work presented here appears to be self-contained and presents results relevant to addressing the hypotheses.

General comments

1. The comparison of the findings reported here to published works demonstrating the occurrence of a mutation in one species is a predictor of its pathogenicity in humans may not be completely valid. From reviewing the cited literature, it appears that this observation is true for very closely related homologs. However, this observation may not be true at larger phylogenetic distances, such as those examined by the authors of this manuscript. It appears that the authors of this manuscript expected to find that human pathogenetic allele would be prevalent in distantly related species. Published results do not seem to suggest this hypothesis. Hence, claiming the results found in this study contradict previous works are not strictly valid because the authors examine much larger phylogenetic distances. I suggest that the authors add a further discussion about how their results differ from that of previous studies addressing differences in phylogenetic distances.

2. Throughout the manuscript, it is suggested that the presence of an allele at a specific site is evidence of the allele’s fitness. The authors should review Kimura’s neutral theory of evolution and address that parallel or convergent evolution is not always indicative of fitness. Zou and Zhang (Mol Biol Evol. , 2015) shows that molecular convergence is often explainable by neutral models.

Specific Comments.

3. The figures that include phylogenies are very hard to read, particularly Figure S9. Consider collapsing clades where a change is not observed.

4. Figure legend 1 claims that arrows are on the figure but none are present. The arrows are included in the supplemental figures but why the distance to flies is included is unclear.

5. Include p-values in the text of the manuscript when you are claiming something is significantly different.

6. Figure 7 does not include a y-axis label.

Experimental design

The research question is well defined, though a more detailed description the initial hypothesis should be given. Please see my first general comment when addressing this. The technical approaches used here appear to be appropriate. However, the methods section is brief and I would like to see the following questions I had about specific lines addressed.

Line 70: How did you constrain the phylogenetic trees?

Line 97: How was the binning of phylogenetic distances done? This really needs to be further explained because your results hinge on the size of these bins.

Line 102: How were these simulations done? Did you use a piece of software?

Line 136: Biased how?

Validity of the findings

Overall, I think the results of this project are sound. However, the manuscript itself needs a significant amount of rewriting. The methods section is brief and the results section does not place findings into a larger context. The authors should take care not to oversell their findings because I don’t think the comparisons to existing literature is valid. If the authors feel that their findings are indeed comparable this needs to be carefully described, addressing differences in phylogenetic distances. Further, the authors need to avoid a selectionist view of molecular evolution. The presence of an allele at a site does not necessarily mean that it confers a fitness advantage, rather its presence may be due to neutral process.

The conclusion section of the manuscripts claims that

Line 240: a variant fixed in humans may be deleterious in non-human species. This was not demonstrated by this study.

The last paragraph from the discussion is just pasted into the conclusions section.

Additional comments

Minor line by line text comments.

Line 38: Include SPFL after you introduce single-position fitness landscape directly after the term.

Line 42: Remove “than in distantly”

Line 52-55: The connection between two exponential distributions and the number of compensatory changes needs to be better explained.

Line 58-61: This is a run-on sentence, consider breaking it up for clarity.

Line 65: It is not clear if the 12 mitochondrial are also from Klink & Bazykin, 2007.

Line 74: Add a citation for the MITOMAP database.

Line 132: What is ‘it’ in the “…these species, it did not share…” sentence?

Line 147: Beyond just reporting thing fold change in phylogenetic distance include some context about what this result means.

Line 183: Change the word order to read “similar to normal human variations than to non-human variants”

Line 207 & 233: Add a sentence about what these differences in phylogenetic distances tells us.

Line 255: Equating phylogenetic clustering and fitness is a false claim.

Reviewer 2 ·

Basic reporting

See comments below

Experimental design

See comments below

Validity of the findings

See comments below

Additional comments

This manuscript is a follow-up paper to Klink and Bazykin 2017 GBE. Whereas the original paper used a careful matching scheme to understand how phylogenetic distance influences the relative frequencies of divergent vs. convergent substitutions, this manuscript applies similar methods to understand homoplastic substitutions to variants that are segregating in human populations and to known human pathogenic variants. In particular, the distribution of phylogenetic distances for these homoplastic substitutions are compared with the distribution of distances for divergent substitutions. The authors find that homoplastic substitutions to variants segregating in humans tend to occur at shorter phylogenetic distance than divergent substitutions and observe a similar pattern for human pathogenic alleles. The authors attribute this pattern to changes in site-specific amino acid preferences over time and also consider a handful of specific mutations in more detail.

Overall I feel this is a well-conducted study. However, I have some suggestions for clarity and issues of interpretation that I feel should be addressed prior to publication.

Major issues:

1. The issue of ascertainment bias in the collection of segregating and pathogenic variants should be addressed in the Discussion. In particular, the database of segregating variants will miss low frequency variants, while the pathogenic variants will miss many mutations that cause lethality during pregnancy (e.g. the pathogenic variants are enriched for mutations whose effects are mild enough to permit viability but severe enough to appear in the medical literature). This means that both types of variants are likely biased in terms of the magnitude of their effects. What impact does this have on the conclusions and interpretation of the study?

2. The current analysis only considers segregating variants and pathogenic alleles that experience at least one divergent and one homoplastic substitution across the phylogeny (line 82). Looking at e.g. Table S2, this can sometimes be a tiny minority of alleles (e.g. for ATP6 only 12 out of 142 polymorphic sites are analyzed). However, this ratio varies greatly between genes and datasets.

If a large fraction of segregating (or pathogenic) variants are never substituted in other species that suggests that the preference against these mutations is consistent over time. This contrasts with the authors conclusion that site-specific preferences change. The authors should discuss the extent to which conditioning on at least one homoplastic substitution influences their conclusions.

3. Throughout, there is a lack of clarity in the language used to describe changes observed in amino acid sequences. For instant in the title it says that “Pathogenic amino acids in mitochondrial proteins more frequently ARISE in lineages closely related to human than in distant lineages.” However, this is confusing because the current manuscript deals with both segregating and fixed differences. Does “arising” mean be produced by mutation or being fixed in the population? Similar ambiguity arises throughout the MS, e.g. line 282 where alleles “emerge independently in species more closely related to H. sapiens.”

I suggest using a clear and consistent terminology, e.g. always refer to “substitutions along lineages” and “alleles segregating” rather than using alleles “arising” or “emerging” to refer to substitutions.


Minor

Line 65 — Please clarify for the reader the degree of overlap between the two datasets for both genes and sequences. Should be clear to reader that these are not independent.

Line 92-93 More detail on pairing procedure. It is not completely clear from the text how the number of samples is determined or whether the divergent and homoplastic substitutions are at the same site or not.

Line 109 “oxydase”->oxidase.

Line 255-257 This is a key sentence, but is hard to parse. Please rephrase.

Line 281-305 Passages here are highly redundant, as if there was an error in editing.

---

## Round 0.2 · Minor Revisions

· Academic Editor

Minor Revisions

Thank you for your careful revisions. There are only a few remaining minor issues that should be addressed, see the comments by reviewer 1. I don't expect that the paper will require further review.

Reviewer 1 ·

Basic reporting

See comments below

Experimental design

See comments below

Validity of the findings

See comments below

Additional comments

I thank the authors for their careful responses and think this version of the manuscript is greatly improved. I feel that all my concerns were proficiently addressed and the changes to the figures significantly improve their clarity. The expanded discussion is especially appreciated and frames this work nicely. My further comments are very minor

1) The first table referred to in the manuscript is S4, so the ordering of materials in the supplemental should be changed.

2) Line 352 includes a typo of the word ‘it’.

Reviewer 2 ·

Basic reporting

See General Comments

Experimental design

See General Comments

Validity of the findings

See General Comments

Additional comments

The revised manuscript is substantially improved. The authors have fully addressed my concerns from the previous round of reviews, and I feel the manuscript is now suitable for publication. I also feel that the added background information in the Introduction and Discussion on patterns in the temporal structure of compensated pathogenic deviations will be helpful to the non-specialist reader. The new analysis and discussion of heteroplasmic-only variants (presumed loss of function) is also interesting, and I found the discussion of the relative frequency of loss of function versus quantitative impairment to be quite helpful. In particular, this provides a reasonable hypothesis to account for the qualitatively different patterns of evolution found in previous studies.

---

## Round 0.3 · accepted · Accept

· Academic Editor

Accept

Thank you for addressing the remaining few issues.